# Recent Advances in Ethylene Gas Detection

**DOI:** 10.3390/ma15175813

**Published:** 2022-08-23

**Authors:** Xiaohu Chen, Ryan Wreyford, Noushin Nasiri

**Affiliations:** NanoTech Laboratory, School of Engineering, Faculty of Science and Engineering, Macquarie University, Sydney, NSW 2109, Australia

**Keywords:** ethylene, gas-sensing technologies, nanomaterials, environmental monitoring, nanostructured gas sensors

## Abstract

The real-time detecting and monitoring of ethylene gas molecules could benefit the agricultural, horticultural and healthcare industries. In this regard, we comprehensively review the current state-of-the-art ethylene gas sensors and detecting technologies, covering from preconcentrator-equipped gas chromatographic systems, Fourier transform infrared technology, photonic crystal fiber-enhanced Raman spectroscopy, surface acoustic wave and photoacoustic sensors, printable optically colorimetric sensor arrays to a wide range of nanostructured chemiresistive gas sensors (including the potentiometric and amperometric-type FET-, CNT- and metal oxide-based sensors). The nanofabrication approaches, working conditions and sensing performance of these sensors/technologies are carefully discussed, and a possible roadmap for the development of ethylene detection in the near future is proposed.

## 1. Introduction

As the simplest alkene [1], ethylene (IUPAC: ethene; molecular formula: C_2_H_4_) has four hydrogen atoms bound to a pair of double-bonded carbon atoms. It has a small kinetic diameter of 4.163 Å with C-H and C=C bond length of 1.096 and 1.335 Å, respectively (Figure 1) [2,3], and a similar specific weight compared to air (1.178 kgm^−3^ versus 1.225 kgm^−3^, respectively, at 15 °C) [4]. Thus, it can freely diffuse in air with an average concentration of ~2 parts per billion (ppb) in atmospheric air [5]. Despite its small size and negligible concentration in nature, it plays significant roles in the environmental and health science sectors. In atmospheric chemistry, ethylene can be produced from automobile exhaust and industrial emissions, which is one of the primary precursors for rapid ozone formation in urban areas [6,7]. Thus, uncontrolled ethylene emissions can result in the increase of ozone concentration at tropospheric or ground level, posing serious health threats [3,8]. Most importantly, it is the well-known natural phytohormone that significantly affects and regulates the root growth [9], seed germination [10], fruit ripening [11,12,13,14], flower senescence [15,16] and the oxidative stress level [17] of plants. For example, most fruits ripen with an ethylene gas concentration in the range of between 0.1 and 1 parts per million (ppm) [13], and flowers usually start to blossom under 0.1 ppm [15,16]. From a biomedical perspective, trace ethylene of around 10 ppb in human breath might be an indicator of lipid peroxidation in lung epithelium compared with a lower exhalation level of ~6.5 ppb for healthy humans [18,19]. It was reported that elderly patients with renal disease exhale elevated ethylene concentrations in a high range of ~150 to 780 ppb compared to the averaged 6.3 ppb of breath ethylene concentration in healthy subjects [19]. Additionally, exogenous ethylene (produced by fruit/vegetable ripening, the incomplete combustion of vehicle fossil fuels, cigarette smoke, etc.) can metabolically form the mutagenic and carcinogenic substance of ethylene oxide in the human body [5,20], Meanwhile, endogenous ethylene can produce toxic ethylene oxide via bacterial production in the gastrointestinal tract and/or systemic formation in the liver [20], causing potential health risks. Therefore, the design and fabrication of highly sensitive, selective, and highly stable gas sensors that can perform the real-time detection of sub-ppm or ppb concentrations of ethylene gas molecules are urgently needed for the agricultural, horticultural and healthcare industries.

The use of advanced nanostructured sensing materials including carbon allotropes (graphene [21], carbon nanotubes (CNTs) [22], graphene oxide (GO) and reduced graphene oxide (rGO) [23]), metal oxides [24,25], and metal–organic frameworks (MOFs) [26,27] has been reported as promising detection techniques for monitoring the concentration of a specific target gas in a complex gas mixture. Although the detection of ethylene gas at a low concentration is challenging due to its small size, lack of polar chemical functionality and limited physiochemical reactivity [28], both nondestructive approaches and chemical reaction strategies are still widely and intensively being explored to achieve this goal. In this work, recent advances in the state-of-the-art technologies for low-concentration ethylene gas detection, their fabrication techniques, working conditions, sensing mechanisms, performance, and corresponding limits of detection are carefully investigated and discussed (Table 1), and a potential roadmap for the future development of highly sensitive and selective ethylene gas-sensing technologies is proposed.

## 2. The State-of-the-Art Ethylene Gas Detection Technologies

### 2.1. Gas Chromatographic System, Non-Dispersive Infrared and Raman Spectroscopy

Gas chromatography (GC) is a commonly used technique to realize the identification and detection of individual gas molecules from mixed-gas compounds [60]. In such a system, different gas molecules can be separated and identified via built-in GC columns due to the different specific retention times of each gas compound. Then, the individual gas molecules can be quantitively detected with a chemiresistive gas sensor at the end of the system by monitoring the generated electrical signals [4,29,30]. In order to reduce the size of a conventional GC system and simplify the sample injection process, Janssen et al. [29] replaced the injection and sampling components with a small preconcentrator. The upgraded GC, namely, large-capacity-on-chip preconcentrator device, enabled the detection of ethylene gas molecules down to 170 ppb, with a theoretical limit of detection (LOD) of 50 ppb. In addition, the preconcentrator filled with the adsorption material of CSII (80–100 mesh in size) showed enhanced separation behavior of ethylene gas from moisture, resulting in an improved cross sensitivity towards ethylene gas measurement. Using a commercially available electrochemical sensor with built-in anti-condensation filter, Zaidi et al. [30] limited the impact of humidity and pushed the practical detection limit of the device down to 20 ppb, with a theoretical LOD of 2.3 ppb. In addition, the modified GC device exhibited excellent repeatability, with negligible fluctuations over five repetitive cycles of measuring ethylene gas at the concentration of 400 ppb, thus suggesting a great potential for on-site ethylene gas monitoring.

The infrared method is another widely used technique for gas detection by analyzing the vibrational or rotational spectra of gas molecules. Gas molecules can absorb the light radiation and generate a band of signature absorption lines centered around a specific wavelength/frequency (peaks). The unique absorption peaks and corresponding absorption intensity (absorbity) can be used for the qualitative and quantitative measurement of gas molecules [61]. Recently, non-dispersive infrared spectroscopy (NDIR) for ethylene detection was studied by Biasio et al. [62], where a characteristic mid-IR absorption peak positioned at 10.54 µm (~950 cm^−1^) was found for ethylene absorption (Figure 2a, inset). The peak intensity increased with increasing ethylene gas concentration (Figure 2a), suggesting a feasible spectroscopic method to monitor ethylene gas using Fourier transform infrared (FTIR) technology. By selecting a band range close to 950 cm^−1^ and using a spectral angle mapper and a cross-correlation filter, a hyperspectral imaging (900–1000 cm^−1^) was obtained for monitoring the regional ethylene distribution [63]. However, water and CO_2_ molecules interfered with the outcomes, as their absorption peaks were very close to ~950 cm^−1^, centering at ~948.26 and ~949.48 cm^−1^, respectively [32]. As a result, the elimination of moisture and CO_2_ prior to IR spectroscopic analysis is highly encouraged for acquiring accurate data.

In another approach, Jochum et al. [31] developed a versatile gas sensor using Raman spectroscopy equipped with hollow-core photonic crystal fibers to simultaneously quantify several gases, including ethylene (Raman shift located at ~1342 cm^−1^), CO_2_, and O_2_ in one single test (Figure 2b). They reported that the Raman intensity of the ethylene peak was accordingly enhanced with increases in the concentration of ethylene in the gas mixture (Figure 2c), demonstrating a quantitative approach for ethylene gas detection in the presence of interfering gases. Despite its low sensitivity among other atmospheric gases, the developed photonic fibers demonstrated a low signal-to-noise ratio (SNR) of 7.2 towards ethylene gas at a high concentration of 100 ppm, suggesting that further studies are required to improve the sensitivity of ethylene gas detection at a low concentration level (sub-ppm).

### 2.2. Photo-/Surface Acoustic Devices and Quartz Crystal Microbalance

Photoacoustic spectroscopy (PAS) is another nondestructive gas detection technique that works based on investigating generated acoustic signals via the piezoelectric effect to measure the concentration of ethylene gas [64]. Traditional PAS equipped with the broadband electric microphone confronts a major challenge of high noise backgrounds from the environment. A recent attempt to replace the conventional microphone and resonant cell with a well-designed quartz tuning fork (QTF) coupled with the micro-resonator (mR), namely, quartz-enhanced PAS (QEPAS), was reported as a promising alternative technology for trace ethylene detection at the ppb level [32,33].

Wang et al. [32] utilized a continuous-wave distributed-feedback quantum cascade laser (QCL) that could emit a 10.5 µm mid-IR at 0 °C to detect ethylene gas under a pressure of 760 Torr. The sensing response continuously and super-linearly increased (R^2^ = 0.9999) with the increase in ethylene gas concentration from 5 to 95 ppm (Figure 3a and inset), demonstrating an extraordinarily quantitative detection capacity of QEPAS towards ethylene gas. The Allan deviation plot, which determines the minimum detection limit (MDL) as a function of the integration time [65], was employed to investigate the minimum detectable concentration of ethylene. An MDL of 560 ppb was reported over 0.1 s, and a ten-fold reduced MDL of 50 ppb could be further achieved for a prolonged integration time of 70 s. In a similar approach, Giglio et al. [33] duplicated a QEPAS-based sensing system for ethylene detection at a lower pressure (120 Torr) and ambient temperature (15 °C), presenting another super linearity of peak signals against the ethylene gas concentration (Figure 3b and inset). By modifying the geometry of the QCL to T-shaped prongs, they pushed the MDL of ethylene gas down to 10 ppb within a short integration time of 10 s. However, the interfering signals generated by the water molecules could challenge the on-site ethylene gas detection, particularly in a humid environment [32]. By injecting different buffer gases, such as xenon, argon, nitrogen, and helium to the acoustic resonator to obtain low background noise, Mohebbifar et al. [35] developed a CO_2_-laser-based photoacoustic spectroscopic gas sensor for ethylene gas detection. A very low noise level of 20 µV was achieved upon the utilization of xenon buffer gas, which was significantly lower than the 30–60 µV nose level obtained by other buffer gases. This low noise level resulted in the highest photoacoustic signal with a low LOD of 3 ppb at an elevated pressure of 765 Torr. However, the utilization of xenon gas can greatly increase operating costs, making it impractical for real-world applications. Further studies are required to replace xenon gas with low-cost buffer gases, such as oxygen and CO_2_, to reduce the cost for real-world ethylene gas detections.

By introducing a thermal-evaporated thin diaphragm of parylene-C (~800 nm) as the acoustic probe, Gong et al. [36] designed a fiber-optic acoustic-based sensor (Figure 3c, inset) capable of simultaneously detecting different gases (C_2_H_2_, CH_4,_ C_2_H_4,_ C_2_H_6_, CO and CO_2_). An excellent linear pattern of the output voltage signal with a high R value of 0.99921 was obtained after exposing to a wide range of ethylene gas concentrations (Figure 3c), with a low LOD of 0.16 ppm. Meanwhile, the sensing system demonstrated a low average deviation of 3.1% for the repeated ethylene gas detection within a gas mixture, suggesting the great potential of this developed technology for reliable ethylene gas detection.

In another approach, Setka et al. [38] fabricated a love-mode surface acoustic wave (L-SAW) sensor for ethylene gas detection by decorating the conductive polypyrrole (PPy)-sensing layer with Au nanoparticles (Figure 3d, inset). A high Au-loading ratio of 1:2 to PPy significantly enhanced the sensing response (defined as the frequency shift) by three-fold due to the promoted absorption of ethylene molecules (Figure 3d). Fast response and recovery time of 81 and 142 s (towards 5 ppm), respectively, and a low LOD of 87 ppb were achieved in dry conditions, suggesting an outstanding ethylene gas detection capacity. However, a dramatic drop in device sensitivity was observed (the ratio between frequency shift change (response) to analyte concentration change) from 270 Hz·ppm^−1^ in dry conditions to 72 Hz·ppm^−1^ at a relative humidity (RH) of 30% [38]. Such a significant decline of sensing performance in wet conditions would hinder its real-world application as a potential sensing technology.

A quartz crystal microbalance (QCM) was modified by Tolentino et al. [39] to detect ethylene gas molecules, which also works based on the piezoelectric effect. Herein, the Ag(I)/polymer composite (AgBF_4_/polyvinylpyrrolidone (PVP)) was creatively added onto the QCM. The Ag iron from silver salt (AgBF_4_) could selectively bind ethylene molecules to form complexes and thus increase the surface mass, resulting in a reduction in the oscillation frequency. A quasi-linear response (frequency shift) against the ethylene gas concentration within a low sub-ppm range (1–7 ppm) and a low LOD of 420 ppb were reported, demonstrating its good potential for ethylene gas detection. However, this QCM-type sensor suffers from several drawbacks including a poor reversibility (with a short lifetime of two runs), lengthy response time (10 min), and high cross-sensitivity towards other gases (e.g., other alkenes due to these molecules can also be bound into complexes with the Ag(I)/polymer composite). Therefore, future work focusing on solving the abovementioned issues would greatly push this silver-functionalized QCM into practical application.

### 2.3. Optical Devices

The cataluminescence (CTL)-based gas sensors are promising new types of chemical transducers that have received great attention in the past three decades due to their excellent performance in analyzing chemicals with high sensitivity, fast response, low background interference, and simple instrumentation [27,66]. Luo et al. [40] reported the first paper-based CTL ethylene gas sensor composed of homogeneously dispersed, Mn-doped SiO_2_ nanoparticles on weighing paper (Figure 4b), which was capable of detecting ppm levels of ethylene gas at room temperature. As presented in Figure 4a, the highest CTL signal was achieved with ethylene gas molecules, demonstrating the high sensitivity and selectivity of this sensing technology. A linear sensing response was observed within the concentration range of 33–6667 ppm (Figure 4c), indicateing a high sensitivity. Furthermore, the luminescent intensity was impressively stable and showed no noticeable degradation over 5 months. Further explorations of LOD enhancement (toward/down to sub-ppm or ppb levels), device portability, and wireless connectivity could lead to the real-time and on-site monitoring of ethylene gas.

In another study, Li et al. [41] developed a colorimetric sensor array for measuring ethylene gas concentration within a complex gas mixture (Figure 4d), demonstrating a high sensitivity down to the sub-ppm level (0.17 ppm). Encapsulated PdCl_2_ in porous silica microspheres (Figure 4e, inset) was used as the sensing material working together with five pH indicators: chlorophenol red, CR; bromocresol purple, BP; bromophenol blue, BB; bromoxylenol blue, BXB; and alizarin, AL. Upon exposure to ethylene gas molecules, the local acidity of the system increased due to the release of HCl from PdCl_2_ microspheres via the Wacker reaction (PdCl_2_ + C_2_H_4_ + H_2_O → Pd + CH_3_CHO + HCl). This resulted in a dramatic color change of the indicators (Figure 4d). A quantitative analysis based on the Euclidean distance algorithm [67] was used to report the gas concentration as a function of the observed color change in red (R), green (G), and blue (B) values (RGB) for each sensor element before and after exposure to the ethylene gas (Figure 4e). The real-world application of the developed sensing technology was investigated by monitoring the different ripening stages of bananas sealed in a zipper bag, revealing that the ethylene gas concentrations derived from the color change (RGB) matched well with the values from FTIR multi-gas analyzer, with an average relative standard deviation of 3.5%. Furthermore, no significant change in the sensing response was observed across exposure to a wide range of humidity parameters (10–90 RH%), suggesting a high stability in wet conditions. Although this colorimetric sensor was designed to be disposable, the viability for commercialization seems economically practicable due to its simple synthesis process and acceptable fabrication cost (20 cents per sensor). However, this sensing technology might not be suitable for detecting ethylene gas at high concentration levels due to the saturation of Wacker reactivity.

### 2.4. Potentiometric Gas Sensors

Potentiometric sensors are a branch of electrochemical sensors that are used to determine the analytical convention of certain components by measuring the potential difference between the working and reference electrodes [68]. Toldra et al. [43] developed a potentiometric gas sensor by using Fe_0_._7_Cr_1_._3_O_3_ nanostructured powders as the working electrodes for ethylene gas detection at 550 °C (Figure 5a,b). In this work, a dense disk-shaped membrane of 8YSZ (8 mol% Y_2_O_3_-stabilized ZrO_2_) layers was utilized as the conducting electrolyte. Screen-printed porous platinum (Pt) was used as the reference electrode due to its high catalytic property and excellent thermal stability. The contacting electrodes were made of gold wires and silver ink, the silver ink was employed to ensure contact between the gold wires and the working electrodes. The fabricated sensor exhibited a good sensing response (defined as the cell voltage; see Figure 5d) towards ethylene gas molecules, indicating a high sensitivity of 0.12 mV·ppm^−1^. Meanwhile, it also exhibited a high selectivity towards ethylene against CO at the operating temperature of 550 °C, suggesting the possibility of ethylene gas detection in the hot exhaust gas streams (containing reducing gases and oxygen molecules) from automobile engines. The cross-sensitivity of ethylene to CO gas molecules was further reduced by decorating the surface of the working electrode with Ti or Al nanoparticles. This could be attributed to the enhanced kinetics of the surface catalytic reactions due to the applied nanoparticle decorations [69]. However, the cross-sensitivity could be significantly affected by the addition of water molecules under humid conditions. To minimize the impact of humidity and to enhance the sensor selectivity towards ethylene gas molecules, the surface of the working electrode (Fe_0_._7_Cr_1_._3_O_3_) was functionalized using Ni nanoparticles (Figure 5c), resulting in significantly higher catalytic activity towards ethylene compared to the CO and water molecules [44,69]. In addition to higher selectivity, the deposition of Ni nanoparticles also improved the sensor response, with a ~300% enhancement in device sensitivity under the same conditions (Figure 5d,e). To improve the conductivity and reduce the cost, the Pt reference electrodes could be replaced by La_0_._9_Sr_0_._1_MnO_3_ (LSM) perovskite-based electrodes due to their higher ionic conductivity [44].

Using 8YSZ as the solid electrolyte as well, Sekhar et al. [45] developed a bilayer electrochemical gas sensor to detect low concentration of ethylene gas molecules at wet conditions (10 RH%). A bilayer of Lanthanum Strontium Chromite (LSC) with a coated rGO–Cu nanocomposite was used as the working electrode. Increasing the operating temperature from 400 to 550 °C would enhance the reaction kinetics and result in faster response time. On the other hand, however, higher operating temperatures caused greater heterogeneous catalysis that could potentially reduce the sensitivity of the device. Therefore, an optimal operating temperature of 500 °C was employed here. In a typical test at 10 RH%, the response monotonically increased in a staircase sequence as the ethylene gas concentration increased from 0 to 100 ppb, and then it demonstrated a near-symmetric sensing dynamic plot as the ethylene gas concentration decreased back to 0 ppb (Figure 5f, black line), showing an outstanding repeatability. However, a 30% drop in the sensing performance was observed when the sensor was tested at a higher humidity condition of 80 RH% (Figure 5f, green line). The Cu nanoparticles used in this device acting as the trapping sites to ethylene gas molecules, prompted the oxidation of ethylene at the electrode/electrolyte interface. The rGO herein provided the conductive pathways for the fast electron transportation (response time of ~80 s). Furthermore, the device exhibited strong interference rejection to NH_3_, NO_2_ and CO gas molecules (Figure 5g). Therefore, working as the solid-state electrochemical cells, these potentiometric device-based gas sensors exhibit several advantages toward ethylene gas detection, such as extraordinary repeatability, low cross-sensitivity, high stability, low LOD (10 ppb), and the capability of working at wet conditions. However, their high operating temperature could hinder their real-world application as practical sensing technologies for the low-concentration detection of ethylene gas.

### 2.5. Field-Effect Transistor-Based Gas Sensors

Field-effect transistor (FET)-based sensors are classic amperometric-type electrochemical devices that work based on the change in their conductivity when exposed to target gas molecules [70]. In general, the sensing elements are immobilized on the semiconductor path connecting to the source (S) and drain (D) electrodes to interact with the analyte gases. The current flow in the sensing element could be modulated within the generated electric field by applying a bias to the gate electrode [71]. Using a simple wire-wound bar-coating method, Khim et al. [72] developed flexible organic field-effect transistors (OFETs) (Figure 6a) consisting of an ultrathin polymer film with a thickness of 2–3 molecular layers (Figure 6b) as highly transparent (up to 90% in visible range) multi-gas sensors. Such facile coating technology was found to be capable of not only fabricating ultra-thin organic films on a large scale (controlled thickness down to 1.5 nm, within an area of 15 × 15 cm^2^) but also minimizing the material waste compared to spin-coating techniques. This is due to the comparatively slow horizontal movement of the coating bar, which offers sufficient time for the wet-coated polymer ink to gradually dry.

The organic sensing film was fabricated by bar-coating the poly[[2,5-bis(2-octyldodecyl)-2,3,5,6tetrahydro-3,6-dioxopyrrolo [3,4-c]pyrrole-1,4-diyl]-alt-[[2,2′-(2,5thiophene) bis-thieno (3,2-b) thiophene-5,5′-diyl]] (DPPT-TT) as a conjugated semiconducting ink material on a polyethylene naphthalate (PEN) substrate, with photo-crosslinked polyimide (PI) as the dielectric layer. Then, Al or Au was thermally evaporated as the source/drain gate-patterned electrodes to form a highly flexible and transparent gas-sensing device (Figure 6a). An atomic force microscopy (AFM) image of a DPPT-TT thin layer with an average thickness of 2.16 nm as presented in Figure 6, demonstrating the outstanding capability of the bar-coating technique for the fabrication of ultra-thin polymer films comprising uniform nanofibrils with molecular-level precision. The high performance of the fabricated sensing device in detecting ethylene gas molecules could be attributed to the ultra-low thickness of the semiconducting layer. As a result, gas molecules could travel to reach the transistor channel faster due to the shortened diffusion path (Figure 6c). However, the sensing performance towards ethylene gas molecules is not practical for real-world applications due to several challenges. For example, the response (G_0_/G_g_, where G_0_ and G_g_ stand for the measured conductance without and with target gas, respectively) dramatically decreased during the repeated test due to the polymer’s poor stability (Figure 6d). In addition, this sensing device suffered from a lack of selectivity towards ethylene, as strong interference was observed from other gas molecules including ammonia and ethanol (Figure 6e,f). Thus, further studies are required to stabilize the polymer semiconductors and to modify the dielectric layer (such as adding GO to the dielectric layer with its enriched functional groups [73]) to enhance the gas-sensing performance and potentially the selectivity of the device towards ethylene gas molecules.

In another study, Besar et al. [46] designed an OFET-based ethylene gas sensor by directly spin-coating poly(3-hexylthiophene-2,5-diyl) (P3HT) as the active semiconductor on a silica substrate (Figure 6g). The fabricated OFET gas sensor demonstrated a 14.6% drain current change (sensing response) upon exposure to 25 ppm of ethylene gas (Figure 6h, blue column). By modifying the sensing material using volatile porogens (N-(tert-butoxy-carbonyloxy)-phthalimide) and micro-size Pd particles (<1 µm), a higher level of porosity was created in the polymer layer (Figure 6g). This structural modification significantly enhanced the sensing response by 206.8% to 30.2% (Figure 6h, purple column). Despite its high sensitivity, the developed OFET-based gas sensor suffered from a slow response dynamic (response time was over 10 min) and poor stability in ambient conditions, hindering its application in the real-time monitoring of ethylene gas. Esser et al. [13] developed a highly selective ethylene gas sensor using single-walled carbon nanotubes (SWCNTs) mixed with a Cu(I) complex, which was drop-casted between gold electrodes on a Si substrate (Figure 7a). The fabricated chemiresistive sensor demonstrated a significant conductivity change (ΔG/G_0_, ΔG = G_g_ − G_0_) upon exposure to ethylene gas molecules (Figure 7b), while no detectable response was observed for pure SWCNT-based FET devices. The high conductivity change was attributed to the interaction between the fluorinated tris(pyrazolyl)borate ligand in Cu(I) complex and ethylene gas molecules, in which the Cu(I) complex could bind ethylene and transform into a Cu(II) complex, thus influencing the conductivity of the intimately mixed CNTs (Figure 7c). The highest sensing response of 1.8% (at a 50 ppm ethylene concentration) was achieved at the optimal ratio of 1:6 for the Cu(I) complex and the carbon atoms of SWCNTs (Figure 7d). In addition, the Cu(I)-complex incorporated SWCNT-FET gas sensor exhibited a remarkably high selectivity towards ethylene gas among a wide range of volatile organic compounds (VOCs) (Figure 7d). This high selectivity was attributed to the inherent recognition of ethylene molecules by the Cu(I) complex. To further improve the sensing performance, 5 wt% cross-linked polystyrene (PS) beads (with a diameter of 0.4–0.6 µm) were added to the mixture, resulting in 1.3–2.2-fold increases in device sensitivity towards ethylene gas. The enhanced sensitivity could be attributed to the greatly increased porosity and specific surface areas of the sensing material due to the added cross-linked polymer beads.

To enhance the sensor’s stability, Hasegawa et al. [47] developed an iridium (Ir)-gated SiC-FET gas sensor for detecting ethylene gas. The fabricated device demonstrated an acceptable repeatability (Figure 7e) and high sensitivity towards ethylene, with the highest sensing response of 1.6 µA (ΔI_D_, the difference of drain current before and after sensing the target gas) for 2.5 ppm of ethylene at the optimal operating temperature of 200 °C (Figure 7f). SiC-FET gas sensors should outperform OFET sensors in terms of stability considering that SiC-based devices can generally work at harsh conditions including high temperatures (SiC-FET, up to 225 °C) [74], high UV radiation hardness (SiC-UV photodetector, subject to the He^+^ irradiation at 600 keV, with fluence range up to 5 × 10^14^ ion·cm^−2^) [75], and extraordinary chemical inertness (SiC-MEMS (micro-electromechanical system), <1.5 nm·h^−1^ etching rate in micromachining solution (HF and KOH mixture)) [76]. Although, the inorganic FET-based ethylene gas sensor might be more practical for on-site implementation at this stage, the OFET type gas sensor for ethylene gas detection is still an attractive topic because of high flexibility and facile machinability of the materials. The poor stability during gas sensing should be one of the major challenges that future research needs to focus on.

### 2.6. CNT-Based Chemiresistive Gas Sensors

Despite the extraordinary electrochemical features of CNTs (carbon nanotubes) used as chemiresistive sensing materials [77], pristine CNTs generally exhibit weak response and low selectivity toward ethylene gas due to their weak interaction with target gas molecules [49,78]. As a result, it is important to functionalize CNTs for achieving high sensitivity and selectivity towards ethylene gas detection. Adjizian et al. [49] introduced a chemiresistive ethylene gas sensor made of boron (B)-doped MWCNTs (multi-walled carbon nanotubes, B:MWCNTs) using chemical vapor deposition (CVD). The sensor could work at ambient conditions and achieve a response (ΔR/R_0_, where ΔR = |R_g_ − R_0_|; R_g_ and R_0_ denote the electric resistance before and after exposure to the analyte gas molecules, respectively) of 0.05% for detecting ethylene gas at 7 ppm. Compared to the responses of 0.14% and 1.63% for CO (2 ppm) and NO_2_ (1 ppm) gases, respectively, the outcomes of ethylene gas detection via B:MWCNTs was not impressive. However, it outperformed the pristine CNTs, which exhibited unnoticeable responses at the same low concentration level toward ethylene gas. Structural defects induced during the CVD process led an improved response compared to the pristine MWCNTs. Density functional calculations suggested that B:MWCNTs is apt to form chemical bonding with the absorbed species. Due to their strong bonding, B:MWCNT-based gas sensors usually experience a prolonged recovery feature (up to 2.5 h at a high temperature of 150 °C) when exposed to water molecules.

In a novel approach, Fong et al. [16] developed a SWCNT/catalytic (single-walled carbon nano tube) mixture to selectively detect ppb levels of ethylene gas under ambient conditions. This solid/liquid-based ethylene gas sensor was prepared by drop-casting the SWCNTs dispersion on top of a glass substrate featuring two gold electrodes, followed by pipetting a drop of liquid mixture (1 µL) containing catalytic palladium on top of the previously drop-casted SWCNTs (Figure 8a,c inset). The sensing mechanism is based on the in situ conversion of Pd(II) to Pd(0) via Wacker oxidation when the SWCNTs/catalytic mixture is exposed to ethylene gas molecules, resulting in charge carrier reduction in the p-type SWCNTs and consequently changes the device conductivity. The fabricated sensing technology exhibited dynamic responses towards a range of ethylene gas concentrations (0.5–50 ppm) (Figure 8b), with a linear response (ΔG/G_0_) against the varied concentration of ethylene gas (Figure 8b inset). A theoretical LOD as low as 15 ppb for ethylene gas detection in air was thus calculated, suggesting a promisingly low ethylene gas limit for monitoring the well-being and ripening states of plants. A negligible decrease in the sensing response was observed after 16 days (stored at 4 °C in the dark), demonstrating an excellent long-term stability for the developed technology. Furthermore, this unique solid/liquid interfaced gas sensor demonstrated remarkable sensitivity towards ethylene among a wide range of alkenes (Figure 8c). This could be attributed to the sidewall-functionalized SWCNTs with 4-pyridyl, and the outstanding electron-donating ability of Pd(0). In addition, the detection and sensing of ethylene gas levels were well-conducted in a flower senescence study, demonstrating the feasibility of using this liquid/solid mixture for real-time and in-field ethylene gas monitoring.

As a nonpolar and small molecule, ethylene gas is difficult to detect directly [4]. However, its hydroformylated and/or ketonized by-products via the Wacker oxidation reaction (e.g., the production of acetaldehyde: C_2_H_4_ + 1/2O_2_ → CH_3_CHO) are relatively active due to their reactive radicals (e.g., carbonyl group) [79] and thus should be easier to be detected. Owing to this inspiration, Ishihara et al. [50] proposed a cascade reaction-type array to sensitively and selectively detecting ethylene gas molecules. The cascade reaction involves a few steps in a specific order: firstly, ethylene molecules were catalyzed to acetaldehyde via the Wacker reaction with the desirable catalyst of Pd–V_2_O_5_–TiO_2_ composites at an optimized temperature of 40 °C. Then, the generated acetaldehyde gas molecules were carried to a polytetrafluoroethylene (PTFE) membrane filter filled with saturated hydroxylamine hydrochloride (HA·HCl) which was spatially isolated from the drop-casted SWCNTs (Figure 8d) for acetaldehyde condensation. Consequently, HCl vapor was emitted due to the chemical reaction of condensed acetaldehyde with HA·HCl (NH_2_OH·HCl + CH_3_CHO → CH_3_CH=NOH + H_2_O + HCl). Finally, the emitted HCl vapor acted as the strong p-dopant to SWCNT-based gas-sensing element and created an apparent electric response (ΔR/R_0_). Figure 8e shows that the sensing response increased with the enhanced ethylene gas concentration level (with an estimated conservative LOD of 0.2 ppm), indicating the high capability of the cascade reaction array in detecting sub-ppm levels of ethylene gas molecules in air. Moreover, the arrayed sensor demonstrated negligible responses to a wide range of VOCs due to their non-reactive nature with the Pd–V_2_O_5_–TiO_2_ and HA·HCl elements, indicating a high ethylene gas molecule selectivity. However, the response of the arrayed sensor could be severely impacted by humidity conditions, which demands extra attention for proper calibration and/or RH regulation prior to ethylene-sensing operation.

### 2.7. Metal Oxide-Based Chemiresistive Gas Sensors

Due to the advantages of high sensitivity, excellent repeatability, simple fabrication, and low cost [80], chemiresistive gas sensors based on nanostructured metal oxides have been intensively investigated over the past few decades for environmental monitoring [81], healthcare [82,83] and medical applications [80]. Using a simple electrochemical deposition method, Sholehah et al. [52] developed a nanostructured, flake-shaped ZnO-Ag sensing material with the flake size and film thickness of 10–20 µm and 18–25 µm, respectively, on a flexible substrate for ethylene gas-sensing application at room temperature. A sensing response of 19.6% (ΔR/R_0_) was achieved towards 50 ppm of ethylene gas, which could be attributed to the surface plasmon resonance (SPR) effect caused by Ag agglomerates formed on the ZnO flakes.

Very recently, Alharbi et al. [54] reported a novel use of LaFeO_3_ (LFO), one of the most common compounds in perovskites, as a highly selective sensing material toward ethylene and acetylene gas detection. Using the sol–gel technique, the LaFeO_3_ nanoparticles were deposited on the top of platinum interdigitated electrodes, followed by a short calcination process (10 min) at high temperatures (500–900 °C) to develop rod-like mesoporous agglomerates for ethylene gas detection (Figure 9a, inset). The ethylene gas-sensing response (R_g_/R_0_) decreased from 65 to 1 by increasing the operating temperature from 150 to 300 °C (Figure 9a), demonstrating an optimal operating temperature of 150 °C for a wide range of ethylene gas concentration (25–3000 ppm). In addition, a significantly fast surface saturation was observed upon the introduction of humidity at 25 RH%, resulting in a sharp drop in sensing response from 65 in a dry atmosphere to 50 under 25 RH% (ethylene concentration of 3000 ppm). Further increasing the RH up to 50% resulted in a negligible impact on the sensing response due to pre-saturated surfaces.

In another approach, Akhir et al. [55] prepared Pd-doped SnO_2_ nanoparticles via a hydrothermal method for ethylene gas detection. It revealed that the doped SnO_2_ (Pd, 3%) nanomaterials could achieve a high response (R_0_/R_g_) of 957.96 towards 100 ppm of ethylene gas molecules at a working temperature of 375 °C, which considerably outperformed the undoped SnO_2_ counterpart in the following two key aspects: (1) The optimal operating temperature of the undoped SnO_2_ samples was found to be as high as 450 °C; (2) under this high operating temperature, the response of undoped samples towards a high concentration level of ethylene (1000 ppm) was only 28.69. In addition, fast-sensing kinetics were observed upon the addition of the Pd dopant, resulting in shorter response and recovery times from 40 and 90 s for undoped samples to less than 10 and 60 s for Pd-doped SnO_2_, respectively. It was suggested that the Pd dopant herein played a vital role in enhancing the performance of ethylene gas detection. Firstly, the incorporation of the Pd dopant created a large amount of adsorption sites on SnO_2_ surface and spontaneously promoted the absorption of gas molecules via the spillover mechanism [55]. As a result, a significant reduction in the depletion region of the SnO_2_ nanoparticles occurred upon exposure to ethylene gas, leading to an enhanced response. Secondly, as a catalyst, surface Pd clusters could facilitate the reaction between the ethylene gas molecules and surface-absorbed oxygen atoms, resulting in rapid response kinetics. Meanwhile, the Pd catalyst lowered the activation energy for the ethylene molecules related oxidation reaction [55], thus led to a lower operating temperature compared to the undoped cases.

In a similar approach, Zhao et al. [28] reported that Pd-loaded SnO_2_ nanoparticles outperformed pristine SnO_2_ in detecting ethylene gas molecules at an optimal operating temperature of 250 °C, which was 100 °C lower than the optimal temperature reported for pristine nanostructured SnO_2_. Although well-featured response dynamic curves and corresponding linearities were recorded for both Pd-SnO_2_ and SnO_2_ (Figure 9b,c), the Pd-SnO_2_ indicated a higher response (R_0_/R_g_) over the concentration range and a greater sensitivity (defined as the slope of response to concentration) of 0.58 ppm^−1^, which was three times greater than that of the pristine SnO_2_ under the same testing conditions. In addition, the Pd-loaded sensor showed a quick response time of 1 s compared to the 7 s for pristine SnO_2_ (Figure 9c), but a slightly prolonged recovery time of 103 s in contrast to the 80 s of the pristine case (explanation to the increased recovery time for the Pd-loaded sensor was not addressed by the authors). In addition, a detectable limit of 50 ppb (Figure 9b, inset) was obtained at the optimal operating temperature of 250 °C, suggesting the potential of Pd-loaded SnO_2_ nanostructures as a promising candidate for the real-time monitoring of ethylene gas molecules.

The observable detection limit of ethylene gas was further pushed down to 10 ppb by Li et al. [56]. A hierarchical porous structure of the Pd-α-Fe_2_O_3_/rGO nanocomposite was developed as an ethylene gas sensor (10 ppb–1000 ppm) in which Pd nanoparticles were homogenously distributed on the surface of flower-like porous α-Fe_2_O_3_ and rGO (Figure 9d, inset). The sensing material demonstrated an outstanding sensing performance towards ethylene gas due to the high specific surface area of the hybrid materials, the catalytic effect of Pd nanoparticles, and the chemically active defect sites provided from rGO. As demonstrated in Figure 9d, the response (R_0_/R_g_) gradually increased with the increasing ethylene gas concentration from 1 to 100 ppm, exhibiting an almost-linear response against the concentration. Meanwhile, it was reported a fast response kinetics of 18 and ~50 s for response and recovery time (1 ppm), respectively. In addition, this porous hierarchical nanocomposite gas sensor showed a high selectivity towards ethylene gas with a high response of 10 compared to the negligible response upon exposure to other VOC gases including ethanol, ammonia, and acetone, thus indicating a feasible design for selective ethylene gas detection. As both Pd nanoparticles and rGO are sensitive to water molecules [85,86], further studies are required to reveal the analytic impacts of humidity conditions on the sensing performance of these nanostructured metal-oxide-based gas sensors and their corresponding stability at various RH levels.

Considering sulfur as a member of chalcogens assigned to Group VIIA, MoS_2_-based nanostructures were also used as chemiresistive gas sensors for ethylene molecule detection. To prepare a practical and field-deployable gas-sensing device for the selective detection of ethylene gas molecules with low cost, Chen et al. [84] bundled exfoliated MoS_2_ nanoflakes with SWCNTs (1.5 nm in diameter) to form a continuous 3D porous copercolation network (Figure 9f, inset). The nanocomposite film was fabricated via the deposition of MoS_2_ and SWCNTs on a flexible polyethylene terephthalate (PET) substrate featuring silver interdigitated electrodes, and then the film was coated by a Cu(I)−tris(mercaptoimidazolyl)borate complex to develop the ethylene gas-sensing device (Cu(I)-MoS_2_-SWCNTs). The conductance rapidly and linearly changed when the device was exposed to ethylene gas in the range of 0.1–1 ppm (Figure 9e), showing the ability for sub-ppm level ethylene gas detection. Moreover, the fabricated sensing technology demonstrated an over ten-times higher response toward ethylene gas (10 ppm), compared to a series of VOCs with higher concentration levels (20–50 ppm) (Figure 9f). This ethylene gas sensor is potentially practical for the agricultural and horticultural industries due to its low power consumption. The combination of three major components in this device, namely, MoS_2_, SWCNTs, and the Cu(I)-pincer complex, is essential to secure such outstanding ethylene sensing performance. In fact, as newly emerging 2D materials from the family of transition metal dichalcogenides (TMDs), the exfoliated MoS_2_ nanoflakes have exhibited great sensing capabilities due to their semi-conducting nature and high specific surface areas [87]. In addition, the metallic SWCNTs can dramatically facilitate the charge mobility and reduce the effective channel length of the percolation network, therefore resulting in fast conductance responses and a comparatively low noise level of 0.005% (compared to the noise levels of 0.02% and 0.2% for MoS_2_-SWCNT and Cu(I)-MoS_2_-composited devices, respectively). Furthermore, the Cu(I)-pincer complex acts as the n-dopant to MoS_2_ for improving the charge-transfer effectiveness, and also performs as the transducer to ethylene gas molecules during the gas-sensing process at the same time. Furthermore, the device exhibited a high stability with no noticeable decline in the sensing performance within a testing timeframe of one month. This high sensing stability could be attributed to the negligible volatility of the organic ligand of Cu(I)-pincer. However, further studies are required to eliminate the impacts of humidity and CO_2_ gas molecules on the sensing performance of the fabricated nanocomposite for practical and real-world implementation.

### 2.8. Dual Metal Oxide-Based Chemiresistive Gas Sensors

As discussed above, SnO_2_ is one of the most used metal oxide semiconductors for gas sensing application, particularly for the detection of VOCs [88]. However, the gas sensing performance of fabricated metal oxide devices is strongly dependent on their surface morphology and synthesis approaches. Using an innovative co-precipitation technique, Leangtanom et al. [57] synthesized highly crystalline CeO_x_-SnO_2_ (CS) nanoparticles with an average particle size of ~5–20 nm (Figure 10a) and different Ce:Sn compositions (0:100, 20:80, 25:75, 33:67, 50:50 and 100:0) as sensitive and selective gas sensing materials for ethylene detection. Under a constant bias of 10 V and an optimal working temperature of 350 °C, the fabricated nanocomposites exhibited an apparent resistance drop upon exposure to ethylene gas molecules. All CS sensors demonstrated positive correlated sensing responses against the variation of ethylene gas concentration (Figure 10b) regardless of their Ce-Sn composition. It revealed that the highest response (R_0_/R_g_) of 5.18 could be obtained for the nanocomposite with 30 wt% of Ce (CS3) (Figure 10b). Meanwhile, CS3 also demonstrated the highest selectivity towards ethylene gas against H_2_, C_2_H_2_, CH_4_, H_2_S, NO_2_, C_2_H_5_OH and C_3_H_6_O (Figure 10c), with a short response time of 12 s to a 10 ppm ethylene gas concentration, suggesting the 30 wt% compositional content of Ce in the CeO_x_-SnO_2_ (CS) nanocomposites can prepare the optimal CS gas sensors for ethylene detection. It is a pity that the studies of recovery time and impacts of humidity, which are also important for assessing the ethylene gas sensing performance, were not discussed herein. The gas sensing device suffers from a high-power consumption resulting from the required operating temperature of 350 °C and high applied bias of 10 V. Power requirements are a hinderance to the deployment of this device for on-site monitoring of ethylene gas detection. 

Using a novel strategy, Jeong et al. [58] successfully designed a unique bilayer of SnO_2_-Cr_2_O_3_ to selectively detect sub-ppm concentrations of ethylene gas. Such bilayer was constituted by a micro-thick SnO_2_ film coated with a nanoscale catalytic Cr_2_O_3_ overlayer (Figure 10d,e). It revealed that the thin, pristine, nanostructured SnO_2_ film with a film thickness of 9 μm exhibited substantially high and non-selective sensing response towards several gases including ethylene, trimethylamine (TMA), NH_3_, ethanol and HCHO. Increasing the film thickness from 9 µm to ~21 µm the responses of the interfering gases were significantly decreased due to the ongoing oxidizations and the limitation of the transport from the top to the sensing layers. Thus, the comparatively stable and small-sized ethylene gas molecules could diffuse towards the lower parts of the sensing layer without significant oxidation, resulting in negligible response changes. The further intensive oxidation of the interfering gases was realized by evaporating the thin layer of catalytic Cr_2_O_3_ on the ~21 µm SnO_2_-sensing region (Figure 10f), where the interfering gases can be excessively oxidized into less- or non-reactive molecules (e.g., H_2_O and CO_2_), thus creating low cross-sensitivity towards ethylene gas detection [58]. An optimal Cr_2_O_3_ layer thickness of ~0.3 µm (Figure 10g) was reported for achieving the utmost enhanced selectivity towards ethylene gas sensing. By using this bilayer-structured ethylene gas sensor, the response (ΔR/R_g_) of ethylene gas (2.5 ppm) was greatly improved to 16.8 (Figure 10h), while the responses of interfering gases with same concentration were significantly decreased below 5. The achieved enhancement is due to the SnO_2_-Cr_2_O_3_ bilayer could efficiently and selectively filter/oxidize the interfering gases into less- or non-reactive species, such as CO_2_ and H_2_O, without sacrificing the transport of the ethylene gas to the sensing region for its comparatively high response. Such dual metal oxide gas sensors in the form of a bilayer exhibited a well-profiled sensing transients (Figure 10i), featuring an almost-linear trend between the response and ethylene gas concentration with an estimated low LOD of 24 ppb (Figure 10h), indicating an intriguing sensing strategy for selectively and sensitively detecting ethylene gas molecules in a complex mixture.

## 3. Summary and Outlook

Detecting and monitoring ethylene gas level is significantly important in the agricultural sector due to its importance to almost all the production chains. The enormous advantages of integrating multidisciplinary technologies should be emphasized and intensively considered, as new materials and emerging strategies can be maximumly contributed to obtain a highly sensitive, selective, reliable, portable and cost-effective gas-sensing technologies with low or negligible power consumption for ethylene gas detection. Here, we have comprehensively reviewed the recent state-of-the-art, cutting-edge technologies/sensors for ethylene gas sensing/detection. Several key parameters, such as fabrication method, working conditions, response dynamic and speed, and LOD, were carefully investigated and compared (Table 1). The strengths and weaknesses of each type of ethylene gas-sensing technologies are summarized below, and potential directions to the roadmap of high-performance ethylene gas detection are also suggested accordingly.

GC, as a mature and widely used technology, could provide robust and reliable ethylene gas measurement with a low LOD of 2.3 ppb and a detectable concentration of a few tens’ ppb. Its major drawbacks, however, are the long and complex operation procedures, cumbersome size, high cost for fabrication, and inevitable moisture inference, which significantly hinder its on-site application for ethylene monitoring. The efforts in the future to modify the currently universal GC of measuring multiple gases into a specialized tool for targeting ethylene gas by developing and using highly sensitive, selective, and good humidity-independent gas sensors can greatly realize the goals for detecting ethylene gas on site. Meanwhile, the attempts to develop the gas sensors that capable of working room temperature are also important due to its great potential to make the future ethylene-GC consume less power, thus to be more portable. Additionally, the ongoing optimization of the preconcentrator-based designs aiming to simplify the sampling process and reduce the gas-sensing duration is also highly encouraged.

FTIR and Raman spectroscopy approaches can detect multiple gases in a single test and offer quick measurements, but their poor spectral resolution and low SNR limit their ethylene gas-sensing capacity toward the trace level. Future work should be considered on improving the SNR and obtaining high spectral resolutions.

Sensing technologies based on the piezoelectric effect, such as acoustic and QCM devices, demonstrate excellent ethylene gas detecting features, such as high accuracy (super linearity of response against the gas concentration), fast and tunable integration times for a desired MDL. The exploration of cost-effective alternatives for laser sources in acoustic setups could greatly promote their commercialization. In addition, the adoption of porous structures in QCM design could potentially enhance their corresponding sensitivity towards ethylene gas.

Due to the ubiquity interference of water molecule to various types of ethylene gas-sensing technologies/sensors, including the GC/FTIR/Raman systems, PAS/QCM devices, and organic/inorganic chemiresistive gas sensors, efforts to design and utilize effective and durable moisture filters are critically important to realize the goals of on-site and accurate ethylene gas detection.

Nanostructured chemiresistive gas sensors will continue taking the lead in the development of high-performance ethylene-sensing technologies, and several constructive tips (as reviewed) are listed below.
Smaller nanostructured dimensions can generally enhance the gas-sensing response.The introduction of a Cu(I) complex and/or noble and transition metal nanoparticles (e.g., Pd, Pt and Ni) can potentially enhance sensitivity and selectivity towards ethylene gas detection simultaneously.Increasing the specific surface area and/or creating highly porous microstructure could greatly boost gas-sensing performance.Secondary energy stimulators, such as bias, UV/visible/IR light, and mechanical energy (e.g., piezoelectric effect), might be considered to co-assist/replace the high operating temperatures for some metal oxide-based gas sensors.The in-depth understanding of the science behind the interaction of gas molecules with sensing elements, such as the dipole fluctuations in ethylene molecules among subtle field effects in graphene, and the development of advanced analytical technologies to trace their corresponding sensing behaviors can pave the way to selectively distinguish decent signals for defining the sensing performance of target analytes, providing more meaningful insights into the gas detection.

## Figures and Tables

**Figure 1 materials-15-05813-f001:**
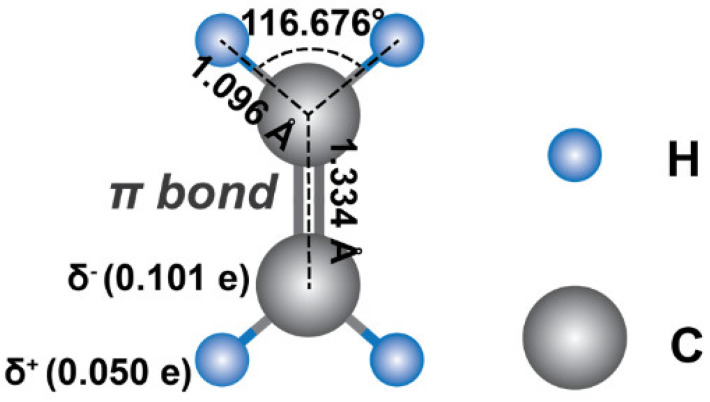
The molecular structure of ethylene.

**Figure 2 materials-15-05813-f002:**
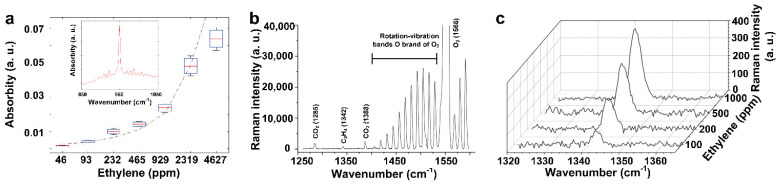
(**a**) Relationship of the absorbity versus the concentration of ethylene gas. Inset: the characteristic absorption spectra, with peaked absorption at the wavenumber of ~949 cm^−1^. Reproduced from Ref. [62] with permission from the IEEE. (**b**) Raman spectrum of 250 ppm of ethylene gas mixed with 20% O_2_ and 500 ppm CO_2_, and (**c**) the spectral intensity of ethylene’s vibration band (1343 cm^−1^) against its concentrations in a mixture of 20% O_2_ and 500 ppm CO_2_. Reproduced from Ref. [31] with permission from the Royal Society of Chemistry.

**Figure 3 materials-15-05813-f003:**
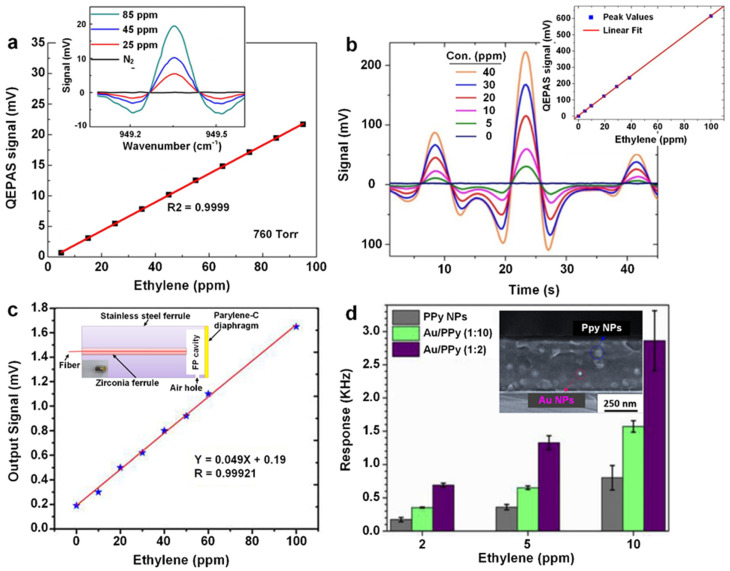
(**a**) Linearity of photoacoustic sensor signals (~949.3 cm^−1^) to the concentration of ethylene gas under 750 Torr. Inset: typical sensing signals of ethylene at the concentrations of 25, 45 and 85 ppm. Reproduced from Ref. [32] with permission from the Optical Society. (**b**) The corresponding signals of the ethylene gas at different concentrations. Inset: the intensity of the dominating signal peak against the concentration of ethylene gas. Reproduced from Ref. [33] with permission from the Optical Society. (**c**) The linear function of the ethylene gas-sensing signal against its concentration. Inset: the schematic and photographic image parylene-C diaphragm-designed, fiber-optic low-frequency sensor head. Reproduced from Ref. [36] with permission from Elsevier. (**d**) The response of three PPy-based L-SAW sensors to ethylene gas at concentrations of 2, 5 and 10 ppm. Inset: side-view of the spin-coated Au/PPy microstructure. Reproduced from Ref. [38] with permission from Elsevier.

**Figure 4 materials-15-05813-f004:**
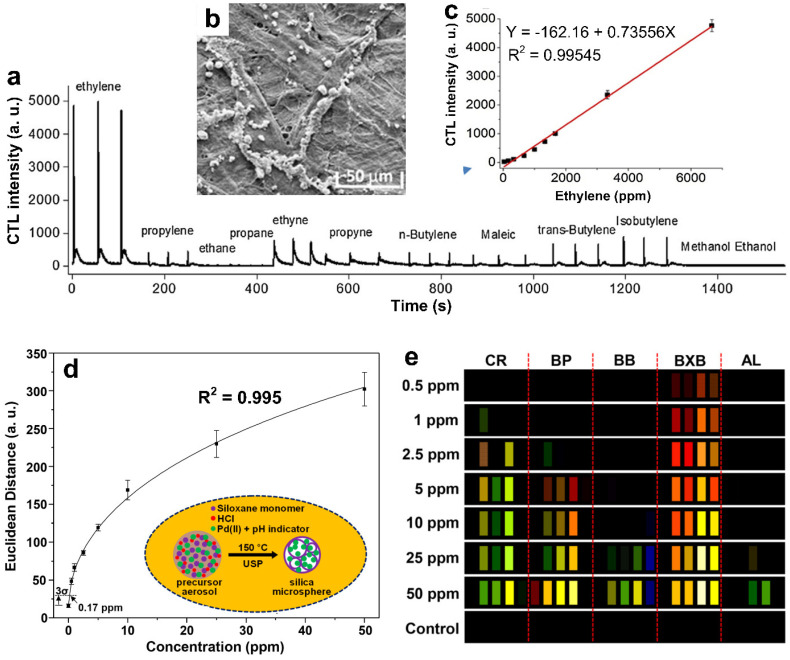
(**a**) CTL signals of a variety of hydrocarbons with a sample concentration of 0.67 vol%. (**b**) SEM image of the nanostructured 0.32 wt% Mn^2+^/SiO_2_ on weighing paper. (**c**) The CTL sensor performance towards ethylene gas detection in terms of the luminescent intensity versus gas concentration. Reproduced from Ref. [40] with permission from Elsevier. (**d**) Calculated response (Euclidean distance) as a function of ethylene gas concentration with an exposure time of 20 min. Inset: the schematic diagram of the colorimetric ink preparation via ultrasonic spray pyrolysis (USP). (**e**) The different color profiles of the colorimetric sensor arrays upon exposure to ethylene gas ranged from 0.5 to 5 ppm for 2 min. Reproduced from Ref. [41] with permission from the American Chemical Society.

**Figure 5 materials-15-05813-f005:**
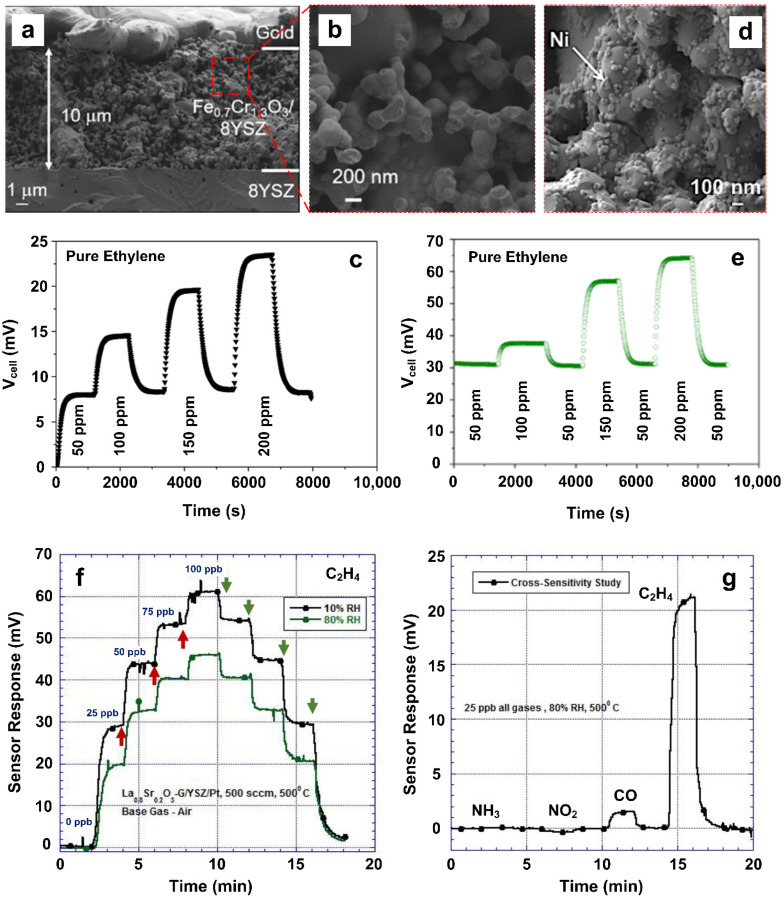
(**a**) Cross-sectional image of the potentiometric ethylene gas sensor; (**b**) magnified SEM image of the rectangular cycle. (**c**) The dynamic response curve of ethylene gas detection at different concentrations. Reproduced from Ref. [43] with permission from the MDPI AG. (**d**) SEM image of the modified potentiometric sensor with incorporated Ni nanoparticles. (**e**) The improved response curve of the modified potentiometric sensor to ethylene gas at varied concentrations. Reproduced from Ref. [44] with permission the American Chemical Society. (**f**) The ethylene-sensing performance of rGO–Cu composited potentiometric gas sensor toward ppb-level ethylene gas at low (10 RH%) and high (80 RH%) wet conditions. The onset of increase and decrease in ethylene concentrations are indicated by the upward (red) and downward (green) arrows accordingly. (**g**) The sensing response toward different 25 ppb gases at 80 RH% condition. Reproduced from Ref. [45] with permission the Electrochemical Society.

**Figure 6 materials-15-05813-f006:**
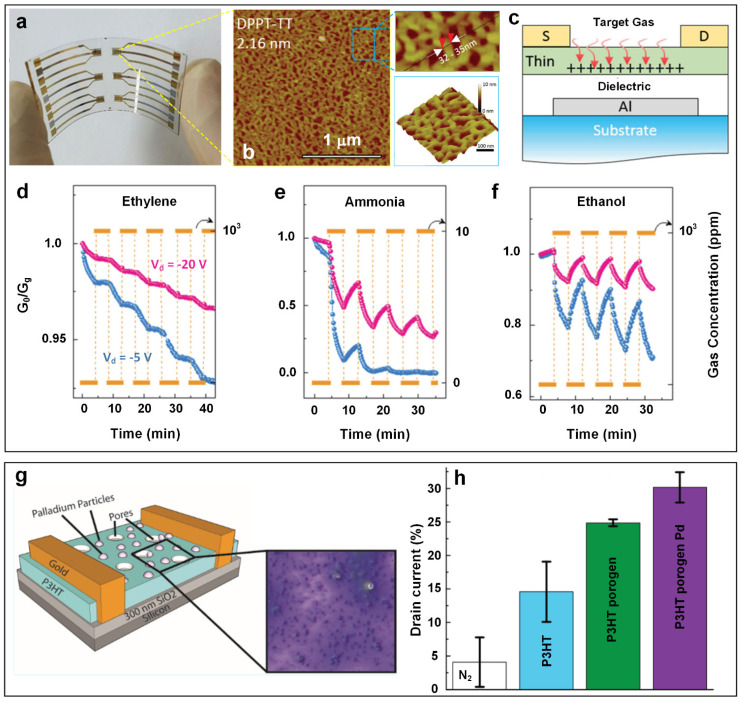
(**a**) Photographic image of the DPPT-TT thin-film-printed flexible gas sensor on transparent PEN substrate and (**b**) the corresponding AFM images of a typical DPPT-TT film. (**c**) Schematic illustration of the thinner OFETs for ethylene gas detection and the corresponding sensing dynamic curves of (**d**) ethylene, (**e**) ammonia, and (**f**) ethanol gases. Reproduced from Ref. [72] with permission from Wiley. (**g**) Schematic of Pd-decorated porous P3HT OFET sensor with the top view as inset. (**h**) The gradually improved response after stepwise modifications. Reproduced from Ref. [46] with permission from the American Chemical Society.

**Figure 7 materials-15-05813-f007:**
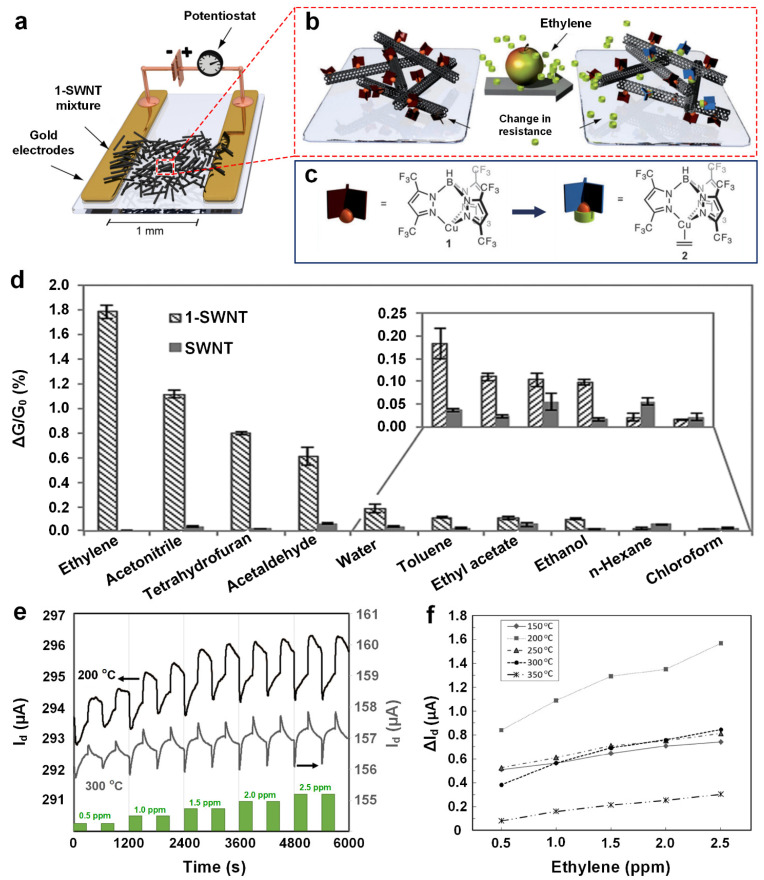
(**a**) Schematics of the Cu(I)-complex-based SWCNT-FET ethylene gas sensor. (**b**) The sensing process and (**c**) the corresponding molecular formulas of the Cu(I) and Cu(II) complex. (**d**) The sensor response towards a wide range of other organic compounds. Reproduced from Ref. [13] with permission from the Wiley-VCH GmbH. (**e**) Typical ethylene gas-sensing dynamic curves of Ir-gated SiC-FET sensor to different ethylene concentrations under 200 and 300 °C. (**f**) Plots of the response versus ethylene concentration at all tested temperatures (150–350 °C). Reproduced from Ref. [47] with permission from the IEEE.

**Figure 8 materials-15-05813-f008:**
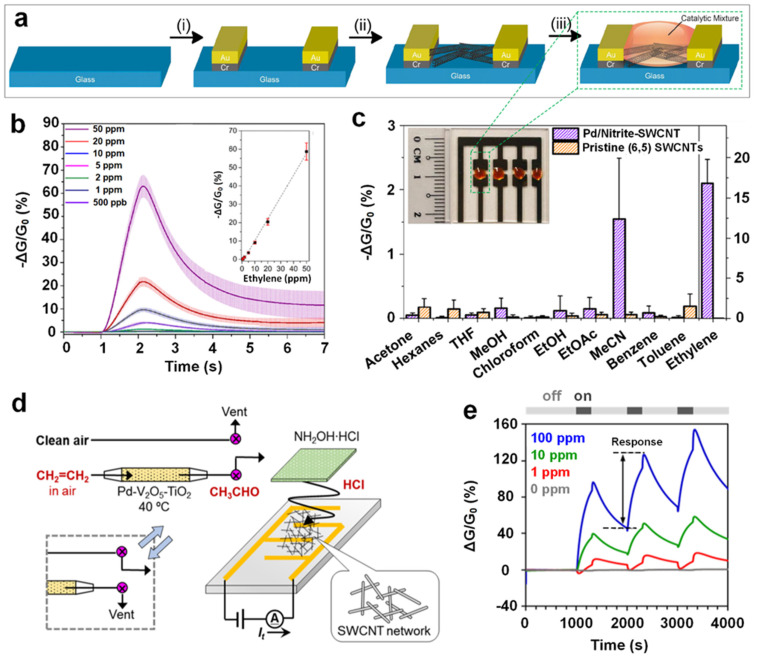
(**a**) Schematic workflow of the liquid/solid device fabrication: (i) electrodes deposition via thermal evaporator, (ii) SWCNTs are deposited between the electrodes via drop casting, and finally (iii) the reaction mixture is added on top of the SWCNTs. (**b**) Sensing dynamic curves and (inset) the plot of response against ethylene gas at varied concentrations. (**c**) The selectivity of the gas sensor toward a wide range of gases including ethylene. Inset: the photograph of SWCNTs with the liquid mixture deposited between Au electrodes on glass. Reproduced from Ref. [16] with permission from the American Chemical Society. (**d**) Pictorial illustration of the sensing system and (**e**) the sensing dynamic curves. Reproduced from Ref. [50] with permission from the American Chemical Society.

**Figure 9 materials-15-05813-f009:**
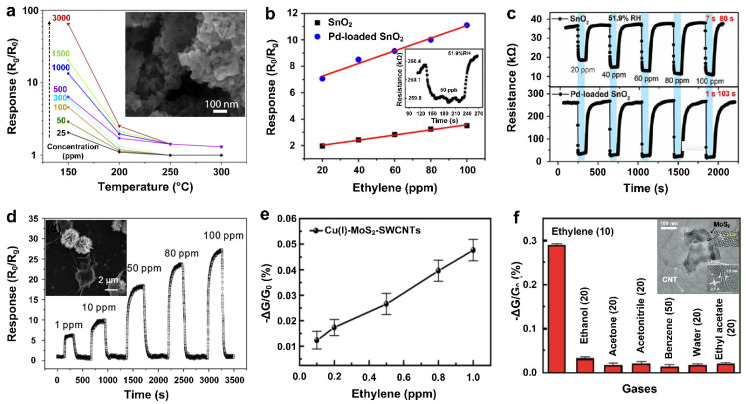
(**a**) Sensing response of nanostructured LaFeO_3_-based sensor toward ethylene gas of various concentrations at different temperatures. Inset: SEM image of nanostructured LaFeO_3_. Reproduced from Ref. [54] with permission from Elsevier. (**b**) Comparison of ethylene gas-sensing response to its concentration for the devices loaded with (bule dots) and without (black squares) Pd nanoparticles. Inset: the sensing curve of 50 ppb ethylene gas when using Pd-SnO_2_ (inset). (**c**) The dynamic curves of SnO_2_-based ethylene gas sensors loaded with (bottom) and without (top) Pd nanoparticles at different concentrations. Reproduced from Ref. [28] with permission from Elsevier. (**d**) The response curve against ethylene gas of different concentration. Inset: SEM image of the Pd/rGO/α-Fe_2_O_3_ nanostructure. Reproduced from Ref. [56] with permission from Elsevier. (**e**) The function of response to ethylene gas concentration and (**f**) the selectivity towards ethylene gas detection among other different VOCs by using the nanocomposite of Cu(I)-MoS_2_-SWCNTs. Inset: the morphology of the MoS_2_-SWCNT nanocomposite under SEM and HRTEM. Reproduced from Ref. [84] with permission from the American Chemical Society.

**Figure 10 materials-15-05813-f010:**
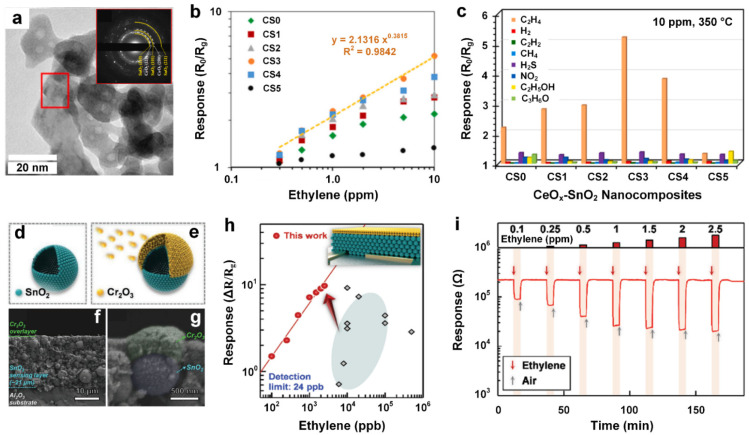
TEM (**a**) and SAED (inset) images of CS3. The relationship of response versus ethylene gas concentration (**b**), as well as selectivity among a ranged of interfering gases (**c**) of all CS sensors. Reproduced from Ref. [57] with permission from Elsevier. The schematic formation of nanostructured SnO_2_ (**d**) and Cr_2_O_3_ overlayer (**e**). Side views of the Cr_2_O_3_-SnO_2_ bilayer (**f**) and the magnification (**g**). Plot of the response against the concentration of ethylene gas (**h**). Inset: the schematic of the Cr_2_O_3_-SnO_2_ bilayer. Dynamic ethylene sensing transients at different concentrations (**i**). Reproduced from Ref. [58] with permission from Wiley.

**Table 1 materials-15-05813-t001:** The state-of-the-art ethylene gas-sensing technologies.

Type	Materials	Fabrication Methods	OT ^$^ (°C)	Interf. Gases ^♦^	RH ^§^ (%)	Con. * (ppm)	Response	Res./Rec. ^+^ (s)	LOD ^&^ (ppb)	On-Site Trial	Refs.
GC	Large-capacity-on-chip preconcentrator device		200	-	-	100	1.9 ^#^; (ΔI)	-	50	-	[29]
	Gas Chromatographic System		45	Ambient air	-	0.4	0.2 V ^#^; (ΔV)	-	2.3	-	[30]
Raman	Hollow-core photonic crystal fibers		RT	O_2_, CO_2_, NH_3_ and N_2_	-	1000	350 ^#^; (I_Raman_) ^a^	600 ^Φ^	100,000	-	[31]
Acoustic	Quartz-enhanced photoacoustic spectroscopy		0	-	-	85	~21 mV; (2f) ^b^	70 ^Ψ^	~50	Apples	[32]
	Quartz-enhanced photoacoustic spectroscopy		15	-	-	100	~250 mV; (2f) ^b^	30 ^Ψ^	7	-	[33]
	Quartz-enhanced photoacoustic spectroscopy		RT	CO_2_	-	35	~40 mV; (2f) ^b^	90 ^Ψ^	8	-	[34]
	CO_2_ laser photoacoustic spectroscopy system		RT	-	0	-	-	-	3	-	[35]
	Fiber-optic low-frequency acoustic sensor		RT	C_2_H_2_, CH_4_, C_2_H_6_, CO and CO_2_	-	30	1.65 mV; (2f) ^b^	-	160	-	[36]
	All-optical photoacoustic system		RT	CH_4_	-	20	0.4 nm ^#^; (RMS) ^c^	-	200	-	[37]
	Au-PPy ^d^	Spin-coating	RT	-	0–50	5	~1.4 KHz; (Δf) ^e^	81/142	87	-	[38]
	AgBF_4_/PVP	Drop-casting	RT	C_6_H_14_, hexane, ethyl acetate ethanol and diethyl ether	-	7	51 Hz·ppm^−1^; (Δf/Con.) ^e^	-	420	Pear, orange and banana	[39]
Optical	Mn:SiO_2_	Surface adsorption	RT	Various alkenes and alcohols	0–0.5	0.67 vol%	~4700; (I_CTL_) ^f^	-	10,000	-	[40]
	Pd(II)-SiO_2_	Ultrasonic spray pyrolysis (USP)	RT	NO_x_, SO_2_, H_2_S and C_2_H_2_	0–90	50	300 ^#^; (RGB) ^g^	1200/-	170	Bananas	[41]
	PCDA/PCDA-SH Liposomes ^h^	Wet chemistry and probe sonication	RT	Air, N_2_ and CO_2_	NG-NA ^Δ^	1000	1.38; ((R/B)/(R_0_/B_0_)) ^i^	-	~600,000	Kiwis	[42]
Potentiometric	Fe_0_._7_Cr_1_._3_O_3_|8YSZ|Pt ^j^	Sol–gel, ball milling, and screen-printing	550	CO	0, 3	200	23.49 mV; (V_cell_)		-	-	[43]
	Ni-Fe_0_._7_Cr_1_._3_O_3_|8YSZ|LSM ^k^	Sol–gel, Ball milling, screen-printing, and drop-casting	550	CO	3	200	65 mV ^#^; (V_cell_)	-	-	-	[44]
	rGO-LSC|YSZ|Pt ^l^	Screen-printing	500	NH_3_, NO_2_ and CO	10–30	0.025	~30 mV; (V_cell_)	80/-	10	-	[45]
Amperometric (FET)	P3HT-Pd ^m^	Spin-coating	RT	Ethyl acetate, methanol and acetone	-	25	30.2%; (ΔI_d_/I_0_) ^n^	-	-	-	[46]
	SiC/Ir	Magnetron sputtering	200	-	-	2.5	1.6 ^#^; (ΔI_d_) ^n^	-	500	Apples	[47]
	MWCNTs	Ink-jet printing and brush coating	RT	-	-	50	18.4%; (ΔR/R_0_)	10/60	-	-	[48]
Amperometric (CNT)	SWCNTs/Cu(I) complex ^o^	Drop-casting	RT	Alkenes, ethanol and acetaldehyde	-	50	1.8% ^#^; (ΔG/G_0_)	-	-	Banana, avocado, apple, pear and orange	[13]
	SWCNTs/Catalytic mixture	Iodonium salt reaction and drop-casting	RT	Variety of VOCs ^p^	40–80	50	~59% ^#^; (ΔG/G_0_)	60 ^φ^/300 ^φ^	15	Lisianthus flowers and carnations	[16]
	B:MWCNTs	CVD and air brushing	RT	-	-	7	0.05%; (ΔR/R_0_)	-	-	-	[49]
	Pd|SWCNTs|HA·HCl ^q^	Impregnation, spin-coating, and drop-casting	40	Variety of VOCs	50	100	~90% ^#^; (ΔG/G_0_)	300 ^φ^/700 ^φ^	200	-	[50]
Amperometric (metal oxide)	ZnO	Electrodeposition and chemical bath deposition	200	-	-	50	2.4%; (ΔR/R_0_)	-	-	-	[51]
	ZnO-Ag	Doping and electrodeposition	RT	-	-	50	19.6%; (ΔR/R_0_)	240/480	-	-	[52]
	ZrO_2_/PTh ^r^	In situ chemical oxidative polymerization	RT	n-hexane, dimethylbutanes, and methyl pentanes	~45	-	9 ^#^; (ΔI/I_0_)	~80/~60	-	-	[53]
	LaFeO_3_	Sol–gel	150	C_2_H_2_, CH_4_, C_2_H_6_, CO, CO_2_ and H_2_	0–50	3000	65; (R_g_/R_0_)	-	-	-	[54]
	Pd:SnO_2_	Hydrothermal	375	-	-	100	957.96; (R_0_/R_g_)	<10/<60	-	-	[55]
	Pd-SnO_2_	Coating	250	-	51.9	100	11.1; (R_0_/R_g_)	1/103	50	Banana, lemon, apple and pear	[28]
	Pd-Fe_2_O_3_/rGO	Calcination, redox reaction, and mechanical shaking	250	Different VOCs	-	10	10; (R_0_/R_g_)	18 ^s^/50 ^s^	10	-	[56]
Amperometric (dual metal oxides)	CeO_x_-SnO_2_	Co-precipitation	350	H_2_, C_2_H_2_, CH_4_, H_2_S, NO_2_, ethanol and acetone	0	10	5.18; (R_0_/R_g_)	12/-	300	-	[57]
	Cr_2_O_3_/SnO_2_	USP, screen-printing, and beam evaporation	350	Trimethylamine, dimethylamine, NH_3_, ethanol, formaldehyde and CO	21, 35	2.5	16.8; (ΔR/R_g_)	6/69	24	Banana, apple, mango, peach, kiwi and blueberry	[58]
	TiO_2_–WO_3_	Sol–gel	250	-	-	200	46.2%; (ΔR/R_0_)	-	8000	Banana, papaya and mango	[59]

^$^ OT: Operating temperature. **^♦^** Interf. Gases: Interfering gases. ^§^ RH: Relative humidity. * Con.: Concentration. ^+^ Resp.: Response time. Rec.: Recovery time. ^&^ LOD: Limit of detection; $LOD=\frac{3RMS_{noise}}{slope}$, where *RMS_noise_* is root-mean-square of measurement noise, *slope* can be obtained from the linear regression fit of sensor response vs. concentration plot. ^#^ Data are not available, estimated value from the graph. ^Φ^ Exposure time of ethylene gas mixture to the Raman sensor. ^Ψ^ Integration time in Allan deviation plot. ^φ^ Time represents the exposure duration to the target gas (ethylene) and air. ^Δ^ NG: Not given; NA: Not affected by RH. ^a^ Intensity of the Raman signal. ^b^ Converted voltage signal (2f) via piezoelectric effect. ^c^ Root-mean-square value of the photoacoustic signal. ^d^ PPy: Polypyrrole. ^e^ Δf: Frequency shift. ^f^ Integrated CTL intensity; CTL: Cataluminescence. ^g^ Euclidean distances (total length of the color difference vector). ^h^ PCDA: 10,12-pentacosadiynoic acid; the thiol-functionalized PCDA is denoted as PCDA-SH. ^i^ R_0_, B_0_, R, and B correspond to the values of the red and blue elements of sensor before and after exposure, respectively. ^j^ YSZ: Y_2_O_3_-stabilized ZrO_2_. ^k^ LSM: La_0_._9_Sr_0_._1_MnO_3_. ^l^ LSC: Lanthanum Strontium Chromite. ^m^ P3HT: Poly(3-hexylthiophene-2,5-diyl). ^n^ ΔId = Change of the drain current. ^o^ Copper complex featuring a fluorinated tris(pyrazolyl)borate ligand. ^p^ VOCs: Volatile organic compounds. ^q^ HA: Hydroxylamine. ^r^ PTh: polythiophene. ^s^ Time was calculated for 1 ppm concentration of ethylene gas.

## Data Availability

Not applicable.

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
