# Peer review of "Recent Advances in Ethylene Gas Detection"

_materials, 2022, doi:10.3390/ma15175813_

Round 1

Reviewer 1 Report

The authors reviewed the recent advances in ethylene gas sensors. This review is well organized with careful analyses of recent papers. These chemical gas sensors have been intensively studied for decades, but most of them have not yet been commercialized. Thus, it should be better to include some critical points to guide readers to an effective approach. Accordingly, the following points should be considered:

1) The actually required target specifications in each application should be clearly summarized. For example, required sensitivity/selectivity in which conditions including temperature/humidity/interfering gas fluctuations and response time. The on-site demonstration is quite difficult for agricultural applications, and thus, some specifications, such as sensitivity to ethylene gas in completely controlled dry air conditions without any interfering gases, are almost no meaning for such applications.

2) Measurement conditions, such as humidity (dry/humid air), interfering gas, and lab/on-site, should be added to Table 1.

3) "Sensing Methods" (e.g. GC, MOX, QCM, Chemiresistive,,,,) should be added to Table 1 in addition to "Materials" and coating "Methods".

4) Electrochemical sensors are one of the standard methods for ethylene gas sensing. Thus, the specifications of these commercial products should be summarized as a reference bench mark, which should be also added to Table 1. The missing/desired specifications beyond these currently available products should be clearly described.

5) In the Summary and Outlook section, there is too much of enphasis on nanostructured chemiresistive gas sensors, which the authors have been studying. Although these approaches seem to have good potential, none of these devices have been commercialized for consumer products, suffering from humidity effects and low reproducibility. Thus, the assessment on these devices is a bit biased and misleading as a fair objective review.

Reviewer 2 Report

I quite like the idea of this manuscript, which is to review recent advances that have been made in the detection of ethylene. Nonetheless, there are a number of issues that combine to detract from my overall enthusiasm for this manuscript, and will need to be addressed before I can recommend publication. These include both several minor issues and a few more major ones. The minor issues include:

1.       In general, this manuscript suffers from a nontrivial of syntax-based errors that make it difficult to focus exclusively on the scientific content of the manuscript. The authors should consider how to best address this issue.

2.       In the abstract, the authors write “the development of ethylene detection in the near further.” They should change the word “further” to “future.”

3.       The first sentence of the introduction indicates that ethylene has a size of 4.163 angstroms. I am confused by this measurement. Is it measuring length? If so, the authors should explain what this length is. It may also be more relevant/useful to be measuring the volume of the molecule, particularly as the review focuses on sensors that are meant to have close-range, selective interactions with the molecule.

4.       Also in the introduction, the authors explain that industrial emissions are one of the “primary precursors for rapid ozone formation.” I presume they mean “ozone depletion” rather than formation, but this needs to be clarified.

5.       In the introduction, the authors talk about patients with renal disease who “exhale elevating ethylene.” Do they mean to write “elevated ethylene concentrations”? This should be clarified and ideally some quantitative information should be added to clarify what is the range of “elevated” concentrations compared to “normal” concentrations.

6.       Also in the introduction, the authors indicate that humans can form ethylene oxide from ethylene via “metabolism process.” This is imprecise and should be clarified/ modified.

7.       At the end of the introduction, the authors state that the variety of methods for ethylene gas detection are “proposed.” This is not accurate, and the word should be changed to “reviewed.”

8.       In Table 1, the abbreviation (Resp./Rec.) needs to be explained the first time that it is used.

9.       Also in general, Table 1 is full of a lot of abbreviations and a tremendous amount of footnotes that makes it overall difficult to understand the content. The authors should consider if/ how they can simplify this table to better convey the required information.

10.   There are several typos in the section that immediately follows Table 1. For example, the authors write “thermotical” instead of “theoretical”; They write “preconcertrator” instead of “preconcentrator,” etc.

11.   In the same section, the authors claim that the researchers got a detection limit “down to 20 ppb,” and then write in parentheses that the LOD of that system was 2.3 ppb. This contradiction needs to be clarified.

12.   On page 5 of the pdf, the authors refer to “the Allan deviation analysis.” This analysis should be briefly explained so that the reader can understand this context and the relevance to the rest of the discussion.

13.   On the top of page 6, the authors explain that a notable drawback to the system they are reviewing is the requirement for Xe gas. They should briefly explain what the use of Xe gas is in the current system, and/ or what other methods might exist to replace the requirement for this costly gas.

14.   On page 7 of the pdf, the authors refer to a significant enhancement “by three folds.” The appropriate phrase is “by three-fold.”

15.   In that same section, the authors refer to a dramatic drop “from 270” to “72.” These numbers need to have units added to them in order to provide the necessary context.

16.   On the same page, the authors talk about the “immobilized Ag iron.” I assume the mean the “Ag ion.”

17.   In the same paragraph, the sentence that starts “Such as the poor reversibility…” is a sentence fragment and needs to be corrected.

18.   On page 8 of the pdf, the authors refer to exposure to “ethylene gas modules.” I assume they mean “ethylene gas molecules.” This should be corrected.

19.   On page 9 of the pdf, I am confused by the phrase “while gold wires, silver ink, and screen-printed porous platinum were used as contacting and reference electrode, accordingly.” This sentence lists three things and then indicates that these were used for two types of electrodes (contacting and reference). It is not at all clear what items were used for which electrode; as such, more clarification is requested.

20.   On page 11 of the pdf, the authors refer to an “optimal operating temperature of 500 oC.” They should clarify on what basis this relatively high temperature is considered “optimal.”

21.   On page 11 of the pdf, in the section agouty FETs, the authors refer to “the created electric filed.” I assume they mean “created electric field.” This should be changed.

22.   On page 12 of the pdf, the sentence that starts “secondly, this sensing device…” is a run-on sentence and needs to be modified.

23.   On page 13 of the pdf, the authors refer to “spin coating.” They should either write “spincoating” or “spin-coating” for clarity.

24.   In that same paragraph, the authors refer to “OFTE.” I assume them mean “OFET.” This should be corrected.

25.   In describing the work of the Swager group on page 13, the authors indicate that the Cu complex changes from “complex 1” to “complex 2.” This is imprecise, and the text should be re-worded to focus on changes in the oxidation state of the copper center.

26.   On page 15, the authors review work in which the researcher’s observed a delta R of 0.03% for the detection of ethylene. The authors should provide enough information to put this value in context; without that context, the value of 0.03% seems too low to be significant for any actual detection application.

27.   In that same discussion, the authors indicate that this value represents an improvement over the pristine MWCNTs. Quantitative information about the response of the pristine nanotubes should be added to this discussion, so that the “improvement” observed upon functionalization can be effectively assessed.

28.   On page 15 of the manuscript, the authors refer to “Thermotical LOD.” This term should be defined and/or modified for clarity.

29.   In that same discussion on page 15, the authors refer to a “wide range of olefins and alkenes.” This is redundant language. The authors should consider using either the term “olefins” or the term “alkenes,” not both.

30.   Near the bottom of page 15, the sentence about the “flower senescence study” is repeated; one of those sentences should be deleted for clarity.

31.   On page 17 of the pdf, the authors write “RH regulation in prior to ethylene sensing operation.” The word “in” is extraneous; the sentence should be modified to read “RU regulation prior to ethylene sensing operations.”

32.   On page 17 of the pdf, line 487 appears to have weird units for measuring the size and thickness of the flakes. The authors should clarify and edit as needed.

33.   Page 17 also has several typos that need to be fixed.

34.   Also, on page 17 the authors indicate that the response decreased “from 65 to 1.” Units need to be provided for these numbers as well as for all numbers used throughout this manuscript (i.e., see page 18, where the response is listed as “957.96,” without units and probably with too many significant figures to accurately represent the scientific reality).

35.   On page 18, the authors refer to 375 degrees C as a “low optimal working temperature.” Some context should be provided to explain why this temperature is considered “low.”

The more major issues include the following:

1.       More significantly, the discussion on page 16 about why ethylene is difficult to detect directly is critical, and should be moved to a more expanded and prominent role near the start of the manuscript. The value of the manuscript is significantly reduced by the fact that the authors jump directly into reviewing how other researchers have detected ethylene, and defer a discussion of how the particular structural features of ethylene inform and influence the various detection strategies reviewed herein.

2.       In addition to adding a more expanded discussion about the structural features of ethylene to the start of the manuscript, the authors should also consider adding expanded information about the different options for detection, the pros and cons of each method, and how they might be used. This general overview would provide a much richer backdrop on which the particular examples of ethylene gas sensing could be assessed.

Reviewer 3 Report

The article by Xiaohu chen et al is devoted to the recent advances for ethylene gas. This review reported the recent state-of-art cutting-edge technologies/sensors for ethylene gas sensing/detection in detail including instrumental analysis method and nanostructured gas sensors. Both of the strengths and weaknesses of these method/sensors are carefully discussed. The review is comprehensive and the work seems to be well done. It provides a possible roadmap for the development of ethylene detection. Overall, the article is certainly of interest, and may be published after a minor revision.

1. Some more references should be added to the list of references to other works on luminescent sensors, for example, DOI: 10.1016/j.jhazmat.2019.121816; 10.1021/acssensors.0c00117.

2. The lower corners of many molecular formulas in the manuscript need to be written in a standardized manner.

Round 2

Reviewer 1 Report

The manuscript seems to be ready for publication although some English editing is required, especially for the revised sentences.